# Web-based self-management support for people with type 2 diabetes (HeLP-Diabetes): randomised controlled trial in English primary care

Elizabeth Murray,[1] Michael Sweeting,[2] Charlotte Dack,[3] Kingshuk Pal,[1] Kerstin Modrow,[1] Mohammed Hudda,[4] Jinshuo Li,[5] Jamie Ross,[1] Ghadah Alkhaldi,[1] Maria Barnard,[6] Andrew Farmer,[7] Susan Michie,[8] Lucy Yardley,[7,9] Carl May,[10] Steve Parrott,[5] Fiona Stevenson,[1] Malcolm Knox,[1] David Patterson[6]

► Prepublication history and additional material are available. To view these files please visit the journal online (http://dx.doi.org/10.1136/bmjopen-2017-016009).

For numbered affiliations see end of article.

**Correspondence to**
Professor Elizabeth Murray; elizabeth.murray@ucl.ac.uk

## ABSTRACT

**Objective** To determine the effectiveness of a web-based self-management programme for people with type 2 diabetes in improving glycaemic control and reducing diabetes-related distress.

**Methods and design** Individually randomised two-arm controlled trial.

**Setting** 21 general practices in England.

**Participants** Adults aged 18 or over with a diagnosis of type 2 diabetes registered with participating general practices.

**Intervention and comparator** Usual care plus either Healthy Living for People with Diabetes (HeLP-Diabetes), an interactive, theoretically informed, web-based self-management programme or a simple, text-based website containing basic information only.

**Outcomes and data collection** Joint primary outcomes were glycated haemoglobin (HbA1c) and diabetes-related distress, measured by the Problem Areas in Diabetes (PAID) scale, collected at 3 and 12 months after randomisation, with 12 months the primary outcome point. Research nurses, blind to allocation collected clinical data; participants completed self-report questionnaires online.

**Analysis** The analysis compared groups as randomised (intention to treat) using a linear mixed effects model, adjusted for baseline data with multiple imputation of missing values.

**Results** Of the 374 participants randomised between September 2013 and December 2014, 185 were allocated to the intervention and 189 to the control. Final (12 month) follow-up data for HbA1c were available for 318 (85%) and for PAID 337 (90%) of participants. Of these, 291 (78%) and 321 (86%) responses were recorded within the predefined window of 10–14 months. Participants in the intervention group had lower HbA1c than those in the control (mean difference −0.24%; 95% CI −0.44 to −0.049; p=0.014). There was no significant overall difference between groups in the mean PAID score (p=0.21), but prespecified subgroup analysis of participants who had been more recently diagnosed with diabetes showed a beneficial impact of the intervention in this group (p = 0.004). There were no reported harms.

**Conclusions** Access to HeLP-Diabetes improved glycaemic control over 12 months.

### Strengths and limitations of this study

► The trial recruited to target and achieved reasonable follow-up; hence, the results for the population of participants are robust (internal validity).

► The two coprimary outcomes reflected the goals of the intervention, namely improving diabetes control and reducing diabetes-related distress.

► However, despite wide inclusion criteria and a deliberately pragmatic design, trial participants were well controlled at baseline, and therefore the extent to which the trial results generalise to the wider population of people with type 2 diabetes is open to discussion (external validity).

**Trial registration number** ISRCTN02123133.

## INTRODUCTION

There is a global epidemic of type 2 diabetes mellitus (T2DM). An estimated 422 million adults, or 10% of the global population, were living with diabetes in 2014 of whom around 90% had type 2 diabetes.[1] Poorly controlled diabetes is associated with premature mortality and a high risk of complications, including cardiovascular disease, nephropathy and retinopathy. The risk of complications can be reduced by good control of glycaemia and cardiovascular risk factors.[2 3] Interventions which improve self-management skills for patients with diabetes can improve health outcomes and reduce healthcare costs[4] and international guidelines support training patients in self-management.[3 5] However, it is not clear how best to support patients in developing such skills, and uptake of diabetes self-management education remains low. In England, despite over 90% of eligible patients

being referred,[6] only 5.3% attended self-management training in 2014–2015.[7]

Poor uptake may be related to the dominant model of structured education, which is group-based sessions, lasting a half or whole day or spread over regular sessions over several weeks.[8] Many patients, such as those who work, those with caring commitments or those who are uncomfortable in groups, may find it difficult to attend.[9 10]

Web-based support for self-management could address some of these barriers, particularly in high-income countries, where levels of web access are high. In the UK, over 80% of households had internet access in 2015, and internet access among older people continues to grow steadily.[11 12] Potential advantages include convenience, anonymity, regular updates and the potential to use video and graphics to present complex information in a format accessible to those with low literacy.[13] Although systematic reviews have confirmed that computer-based interventions can improve health outcomes in diabetes,[14] not all such interventions have a beneficial impact, with meta-analyses showing substantial heterogeneity related to widely differing interventions, including in the use of theory to develop the intervention,[15] outcomes[14 16] and the duration of follow-up, with most trials having relatively short follow-up (less than 12 months).[14] This is the first UK-based trial of a comprehensive, web-based self-management support programme for people with type 2 diabetes.

This trial assessed the effects of a web-based self-management programme, called Healthy Living for People with Diabetes (HeLP-Diabetes), on glycated haemoglobin (HbA1c) and diabetes-related distress over 12 months.

## METHODS
### Trial design and participants
Multicentre, two-arm individually randomised controlled trial in 21 general practices in England with a mix of urban, suburban and rural practices. Practices were required to have two nurses—one to facilitate access to the intervention, and one to collect data.

### Recruitment
Standard opt-in recruitment procedures were followed. Each practice had a register of patients with T2DM. The electronic medical record of every patient on this register was reviewed to screen out ineligible patients, and the remainder were sent a letter from their general practitioner (GP), inviting them to participate in the study. Eligible participants were adults, aged 18 or over, with T2DM, registered with participating general practices. Patients were excluded if they were unable to provide informed consent; unable to use a computer due to severe mental or physical impairment; had insufficient spoken or written English to use the intervention (operationalised as unable to consult without an interpreter); were terminally ill with less than 12 months life expectancy; or were currently participating in a trial of an

alternative self-management programme. Participants were not required to have home internet access or prior experience of using the internet to participate. Participants with previous or current experience of self-management education were eligible to participate. Recruitment took place between September 2013 and December 2014. The trial protocol was submitted for publication in June 2014.[17] There were no changes to the methods after the protocol was agreed and the start of the trial. Ethical approval was obtained from Camden and Islington National Research Ethics Service committee, reference 12/LO/1571.

### Patient involvement
Patients were involved in all stages of the study, including contributing to the original application for funding as coinvestigators; substantive and ongoing contribution to intervention development; contributing to the trial design, including the decision to have two coprimary outcomes; active membership of the Trial Steering Committee and Trial Management Group and contributing to the writing of this paper. This last role is recognised through coauthorship (MK).

### Randomisation and blinding
Randomisation marked the point of study entry. It was performed centrally (independently of the trial team), after written informed consent was obtained and all baseline data were completed, using a web-based randomisation system, at the level of the individual participant. Randomisation was conducted in a 1:1 ratio using random permuted blocks of sizes 2, 4 and 6, stratified by recruitment centre. Participants were informed the trial compared two forms of web-based support, and were blinded as to which was the intervention and which the comparator. Nurses who offered facilitation for the intervention could not be blinded, but were asked not to discuss details of allocation with the nurses who gathered follow-up data. The research team obtaining and analysing data from participants were blind to allocation.

### Intervention
The intervention consisted of facilitated access to HeLP-Diabetes. Facilitation consisted of an introductory training session with the practice nurse. In this appointment, patients were were shown how to log on, set a user name and password and introduced to the content of the website. HeLP-Diabetes was a theoretically informed web-based programme whose overall goals were to improve health outcomes and reduce diabetes-related distress.[18] Overall, content was guided by the Corbin and Strauss model of managing a long-term condition which posits that patients must undertake medical, emotional and role management.[19] It was developed using participatory design principles, with substantial input from users, defined as patients with T2DM and health professionals caring for such patients. All content was evidence-based, drawing on evidence on management of diabetes,

promoting behaviour change and emotional well-being and maximising usability and engagement. Content was designed to be accessible to people with a wide range of literacy and health literacy skills, with all essential content provided in both video and text. There were information sections on diabetes, how diabetes is treated, possible complications of diabetes, possible impacts of diabetes on relationships at home and at work, dealing with unusual situations like parties, holidays, travelling or shift work and what lifestyle modifications will improve health. There were sections addressing skills and behaviour change, including behaviour change modules on eating healthily, losing weight, being more physically active, smoking cessation, moderating alcohol consumption, managing medicines, glycaemic control and blood pressure control. Users could set the programme to send themselves reminder text messages or emails, and could specify the content and frequency of such reminders. The third strand of components focused on emotional well-being with self-help tools based on cognitive behavioural therapy and mindfulness. There were multiple personal stories (used with license from health talk online), and a moderated forum. Participants were free to use the programme as much or as little as they chose. Engagement with the programme was promoted through regular newsletters, emails and short message service containing updates on latest diabetes-related research or practice, seasonally relevant advice (eg, fasting during Ramadan, benefits of 'influenza' vaccinations), and links to specific relevant parts of the programme. Two or three prompts were sent each month, although users could opt-out of receiving them. Further details are provided in online supplementary appendix 1.

### Comparator

From a National Health Service (NHS) perspective, the important research question was whether the proposed intervention could improve health outcomes when compared with current practice. However, to improve acceptability to participants and to maintain blinding, all participants had access to a website. Participants in the control arm were given access to a simple information website, based on the information available on the website of the main UK diabetes charity (Diabetes UK) or National Health Service patient information website (NHS Choices). They received the same initial facilitation meeting as participants in the intervention group, in which they were shown how to log on, set a user name and password and how to use the website.

### Outcomes and outcome measures
#### Primary outcomes

The outcomes reflected the dual goals of improving health outcomes and reducing diabetes-related distress. The two joint primary outcomes were HbA1c and diabetes-related distress, measured by the Problem Areas in Diabetes (PAID) scale, both at 12 months postrandomisation. PAID has 20 items focusing on areas that cause difficulty for people living with diabetes, including social situations, food, friends and family, diabetes treatment, relationships with healthcare professionals and social support.[20] PAID scores range from 0 to 100, with higher scores indicating more distress. A score of 40 or more indicates significant distress, and around 40% of patients with diabetes experience significant distress.[21]

#### Secondary outcomes

Clinical secondary outcomes included systolic and diastolic blood pressure, body mass index, total cholesterol and HDL (not fasting), and completion of the 'nine essential processes' for effective management of diabetes, mandated by NHS England (weight, blood pressure, smoking status, measurement of serum creatinine, cholesterol and HbA1c, urinary albumin and assessment of eyes and feet) within the previous 12 months.[3] Patient-reported outcomes included depression and anxiety, measured using the Hospital Anxiety and Depression Scale (HADS),[22] diabetes-related self-efficacy measured using the Diabetes Management Self-Efficacy Scale (DMSES),[23] and satisfaction with treatment, measured using the Diabetes Satisfaction with Treatment Questionnaire status and change version (DTSQs and DTSQc).[24]

### Data collection

Data were collected at baseline, 3 and 12 months, with 12 months the primary endpoint. Patient-reported data were collected using online questionnaires emailed to participants. Clinical outcomes were collected by nurses in participating practices. Participants were asked to complete their online questionnaires before visiting the nurse for clinical measurements and blood tests. Blood samples were analysed at the local NHS laboratory used by participating practices for routine clinical analyses. Data on completion of the 'nine essential processes' were collected from the GP record for the 12 months prior to randomisation and the 12 months after randomisation at the 12-month follow-up point to avoid triggering behaviour change among the study nurses. Use of the intervention was recorded automatically using bespoke software that recorded the date, and time of each page visited. A new log-in to the intervention was defined as any page that was accessed 30 min or more after the last accessed page.

### Sample size calculation

Our original sample size calculation was that randomising 350 participants with 85% follow-up would provide 90% power at the 5% level of significance to detect a 0.25% difference in HbA1c and a 4.0 point difference in PAID score at 12 months postrandomisation between the randomised groups.[25 26] Since HbA1c and PAID were joint primary outcomes measuring different aspects of T2DM, both were tested at a 5% significance level.

### Analysis

The analysis followed a prespecified analysis plan, based on comparing the groups as randomised (intention-to-treat).

The analysis plan was approved by the Trial Steering Committee before unblinding and uploaded to the trial website (https://www.ucl.ac.uk/pcph/research-groups-themes/ehealth/projects/projects/helpdiabetesrct). Only HbA1c and PAID measured within 10–14 months window period following randomisation was used in the primary analysis with missing 12-month outcomes multiply imputed using baseline and other outcome data (eg, 3 month data and final follow-up data collected outside the 10–14 months window). Further information on the imputation method is given in online supplementary appendix 2.

A linear mixed effects model with random centre effects was used to analyse each of the primary outcomes separately, adjusting for the baseline level of the outcome, age, gender, previous participation in other self-management programmes, pre-existing cardiovascular disease and time since diagnosis of diabetes. Secondary outcome measures were analysed similarly using generalised linear mixed models, with a normal residual error structure for continuous outcomes and a logit link for the binary outcome 'completion of nine essential processes'. Prespecified subgroup analysis for the coprimary outcomes was undertaken by baseline glycaemic control (HbA1c outcome only), baseline PAID (PAID outcome only) and duration of diabetes, treating all potential effect modifiers as continuous. The interaction between randomised group and each effect modifier was included in the model separately and assessed using a Wald test.

Use of the intervention was investigated as a mediator for efficacy, using instrumental variable methods, with randomisation as the instrument (online supplementary figure 1).[27 28]

Potential contamination was monitored by recording participants with similar family names and identifying those with the same addresses. Where this occurred, it was dealt with in the analysis by reporting the extent and undertaking a sensitivity analysis excluding these individuals.

A number of other sensitivity analyses were performed to assess the robustness of the primary analyses: (1) performing two complete case analyses disregarding outcomes measured outside 10–14 months and 11–13 months postrandomisation; (2) repeating the analysis using multiple imputation of baseline covariates only; (3) fitting linear models excluding centre random effects and (4) fitting an unadjusted model using only outcome measured in 10–14 months postrandomisation.

The Trial Steering Committee (TSC) took on the role of the data monitoring committee. Trial registration ISRCTN02123133.

## RESULTS

Recruitment took place between September 2013 and December 2014. An initial 421 patients consented to participate, but of these 47 did not fully complete their baseline questionnaires and were therefore not randomised and did not enter the study. A total of 374 participants were randomised, of whom 86% (n=321) provided data on PAID and 78% (n=291) had HbA1c measured within 10 to 14 months of randomisation. Additional final outcome data, obtained outside the 10–14 month predefined window, were available for a further 27 participants for HbA1c and 16 participants for PAID (figure 1). Data obtained outside the 10–14 months window were not used directly in the primary analysis, but were entered into the imputation model (online supplementary table 1).

### Baseline characteristics
Baseline demographic and clinical characteristics are shown in table 1. The mean age was nearly 65 years, over two-thirds (n=258, 69%) were men and most were White British (n=300, 80%). Nearly all (n=370, 99%) had a computer with access to the internet at home and just over half (n=210, 56%) rated themselves as experienced computer users. Around one-third (n=134; 36%) had been diagnosed for less than 5 years, with a further third (n=115, 31%) having been diagnosed between 5 and 9 years ago. Overall, this was a population with well-controlled diabetes at baseline (mean HbA1c was 7.3% (56 mmol/mol)) and low levels of distress (mean PAID=19).

### Primary outcomes
At 12 months the primary analysis showed a significant difference in change in HbA1c between the randomised groups with participants in the HeLP-Diabetes group having a lower HbA1c than those in the control group (mean difference=−0.24%; 95% CIs −0.44 to −0.049, p=0.014) (table 2, figure 2). There was no difference in change in PAID scores between the groups at 12 months (mean difference −1.5; 95% CI −3.9 to 0.9, p=0.209), though both groups showed a decrease in PAID over the follow-up of the trial (table 2, figure 3).

### Secondary outcomes
There was no difference in secondary outcomes at 12 months, with the possible exception of systolic blood pressure, which decreased more in the intervention group than in the control group (p=0.010) (table 2); though the result was not statistically significant after correction for multiple testing of secondary outcomes. There were no significant differences between groups on any of the outcome measures among individuals who completed 3 month outcomes (online supplementary table 2). No adverse effects or events were recorded during follow-up.

### Usage data
The mean number of log-ins was significantly higher in the intervention group than the control group (18.7 vs 4.8, p=0.0001), as was the mean number of pages visited per log-in (10.5 vs 7.7, p<0.0001) and the mean number of days in which the website was accessed (10.1 vs 3.3, p<0.0001) (table 3). The causal analyses estimated that for a 'high-usage' population (those with usage greater than or equal to the median of 4 days) the HeLP-Diabetes

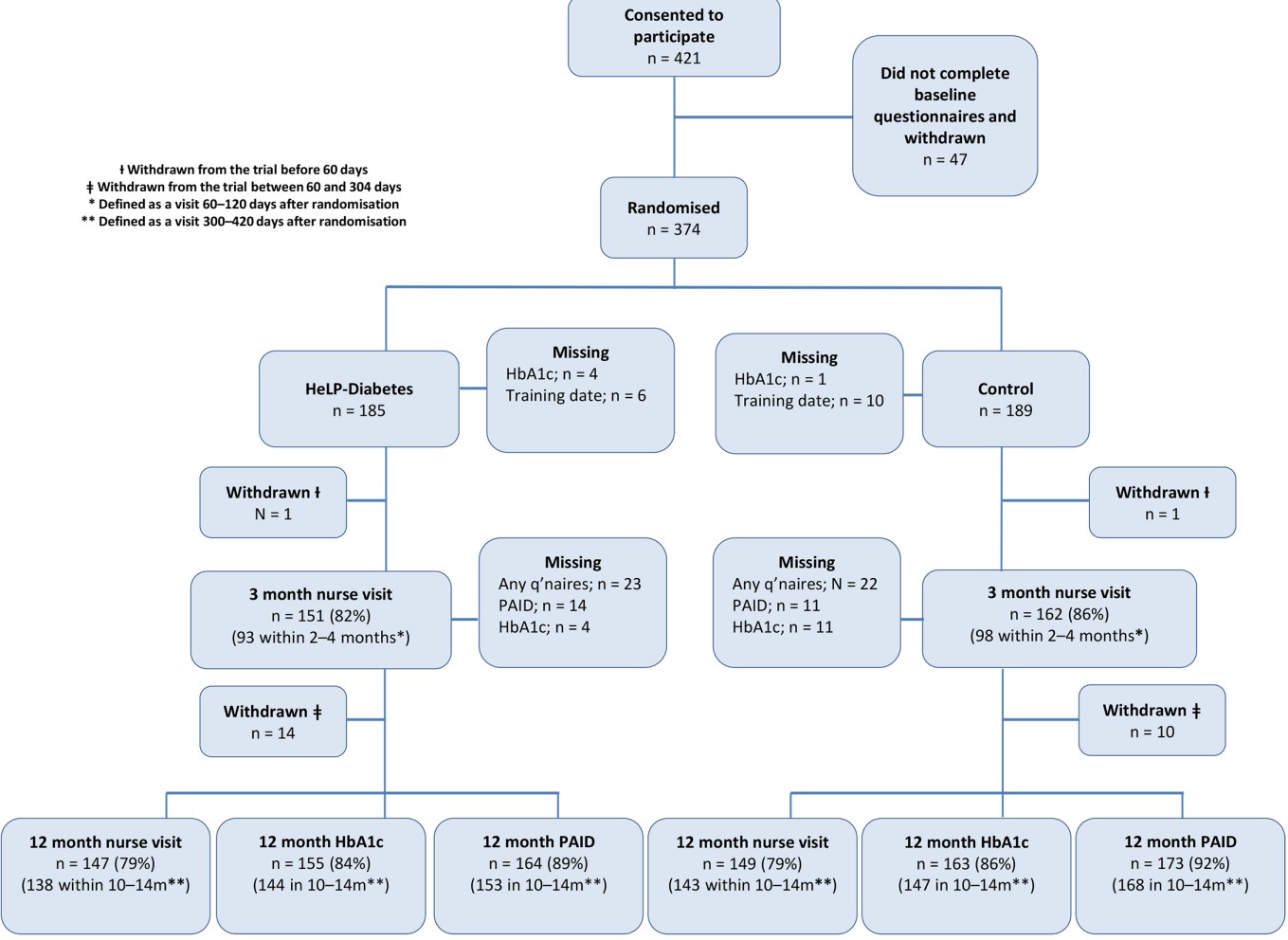

**Figure 1** CONSORT diagram showing patient flow through the HeLP-Diabetes randomised controlled trial. HbA1c, glycated haemoglobin; HeLP-Diabetes, Healthy Living for People with Diabetes; PAID, Problem Areas in Diabetes.

intervention could on average reduce HbA1c by −0.44% (95% CI −0.81 to −0.06) and PAID by −2.8 (95% CI −7.2 to 1.7) over 12 months (online supplementary figures 2 and 3). The mean usage in the 'high-usage' group was 18 days. It should be noted that the usage data presented do not include the initial facilitation visit. There was a technical error in the software which led to usage data not being collected before 1 January 2014. At this point 16 participants had been randomised (seven to intervention, nine to control). For these 16 participants, the usage data are not based on a full year, but for all other participants, data are summarised for the 12 months postrandomisation.

### Sensitivity analyses

The findings from the sensitivity analyses, including a complete-case analysis, were similar to the main analysis (online supplementary table 3). Participants who were missing 12 month HbA1c had significantly higher mean baseline HbA1c measures (7.9% vs 7.1%, p<0.001) leading to higher imputed HbA1c at 12 months in the non-completers and a greater mean difference between the randomised groups than from complete case analyses (online supplementary figure 4, supplementary table 3).

### Subgroup analyses

Prespecified subgroup analyses showed that there was no evidence of baseline measures of HbA1c or PAID being effect modifiers for the mean difference between the groups. There was strong statistical evidence (interaction p=0.004) to suggest that the duration of diabetes acted as an effect modifier, with those who had been diagnosed more recently showing more of a reduction in PAID than those who had been diagnosed for longer periods of time. Duration of diabetes had no effect on change in HbA1c (online supplementary table 4).

### Harms

There were no reported harms in either group.

### DISCUSSION

In this first UK-based trial of a web-based self-management programme for people with T2DM, participants randomised to HeLP-Diabetes demonstrated improved glycaemic control at 12 months compared with those randomised to a simple information website. This improvement appears robust across all prespecified sensitivity

**Table 1** Descriptive statistics of baseline variables by randomised group

| | HeLP-Diabetes n=185 | Control n=189 | N missing |
|---|---|---|---|
| Age at randomisation (years) | 64.9 (9.5) | 64.7 (9.1) | 0 |
| Male sex, n (%) | 127 (69%) | 131 (69%) | 0 |
| Ethnicity, n (%) | | | 1 |
| White English, Welsh, Scottish, Northern Irish, British | 151 (82%) | 149 (79%) | |
| Indian | 12 (6%) | 8 (4%) | |
| Other | 21 (11%) | 31 (16%) | |
| Experience with computers, n (%) | | | 0 |
| None | 5 (3%) | 4 (2%) | |
| Basic | 75 (41%) | 80 (42%) | |
| Experienced | 105 (57%) | 105 (56%) | |
| Smoking status, n (%) | | | 0 |
| Current smoker | 14 (8%) | 14 (7%) | |
| Former smoker | 94 (51%) | 86 (46%) | |
| Never smoker | 77 (42%) | 89 (47%) | |
| Time since diagnosis (years), n (%) | | | 4 |
| 0–4 years | 70 (38%) | 64 (34%) | |
| 5–9 years | 55 (30%) | 60 (32%) | |
| 10–14 years | 40 (22%) | 40 (21%) | |
| 15+ years | 18 (10%) | 23 (12%) | |
| Attending any other self-management class, n (%) | 4 (2%) | 4 (2%) | 0 |
| Clinical measures | | | |
| Systolic blood pressure (mm Hg) | 135 (17) | 135 (17) | 0 |
| Diastolic blood pressure (mm Hg) | 78 (11) | 77 (10) | 0 |
| Total cholesterol (mmol/L) | 4.11 (1.03) | 4.18 (0.98) | 2 |
| HDL-C (mmol/L) | 1.24 (0.31) | 1.25 (0.36) | 12 |
| Total cholesterol/HDL cholesterol ratio | 3.43 (1.09) | 3.52 (1.03) | 13 |
| HbA1c (%) | 7.26 (1.25) | 7.35 (1.37) | 5 |
| HbA1c (mmol/mol) | 56 (14) | 57 (15) | 5 |
| Body mass index (kg/m$^2$) | 30.1 (5.3) | 29.6 (5.2) | 2 |
| Questionnaires/scores | | | |
| PAID (0–100) | 18.1 (17.1) | 19.9 (19.9) | 0 |
| HADS (0–42) | 9.28 (6.47) | 9.12 (7.52) | 0 |
| Anxiety scale (0–21) | 4.92 (3.70) | 5.21 (4.20) | 0 |
| Depression scale (0–21) | 4.36 (3.48) | 3.91 (3.73) | 0 |
| DMSES (0–150) | 98.6 (33.9) | 103.7 (32.4) | 0 |
| DTSQ (0–48) | 32.1 (7.3) | 32.0 (7.2) | 0 |
| Completion of nine essential processes in previous 12 months, n (%) | 97 (64%) | 96 (62%) | 69 |

DMSES, Diabetes Management Self-Efficacy Scale; DTSQ, Diabetes Treatment Satisfaction Questionnaire; HADS, Hospital Anxiety and Depression Scale; HbA1c, glycated haemoglobin; HDL, high-density lipoprotein; HeLP-Diabetes, Healthy Living for People with Diabetes; PAID, Problem Areas in Diabetes.

analyses, and was not dependent on duration of diabetes, baseline glycaemic levels or level of diabetes-related distress. Each 1% reduction in HbA1c is associated with a risk reduction of 21% for deaths related to diabetes and a 37% risk reduction for microvascular complications.[26]

A reduction in HbA1c of 0.24% across a population level could translate into considerable population benefit, particularly as this web-based intervention could be delivered at low-cost and at scale across the UK. Moreover, in contrast to group-based education, where the effects

**Table 2** Twelve-month outcomes, adjusted for relevant baseline outcome, age, sex, current (baseline) participation in other self-management programmes, pre-existing cardiovascular disease and duration of diabetes

| | HeLP-Diabetes | | Control | | HeLP-Diabetes vs Control | |
|---|---|---|---|---|---|---|
| | Baseline | Change from baseline to 12 months | Baseline | Change from baseline to 12 months | Mean difference (95% CI) | p Value |
| **Primary outcomes** | | | | | | |
| HbA1c, (%) | 7.3 (0.1) | –0.08 (0.07) | 7.3 (0.1) | 0.16 (0.07) | –0.24 (–0.44 to -0.05) | 0.014 |
| HbA1c, mmol/mol | 56.3 (1.1) | –0.8 (0.8) | 56.8 (1.1) | 1.8 (0.8) | –2.6 (–4.8 to-0.5) | 0.014 |
| PAID | 18.2 (1.3) | –4.1 (0.9) | 19.8 (1.3) | –2.5 (0.9) | –1.5 (–3.9 to 0.9) | 0.209 |
| **Secondary outcomes** | | | | | | |
| Systolic blood pressure, mm Hg | 134.7 (1.5) | –4.2 (1.4) | 134.9 (1.5) | –0.5 (1.4) | –3.8 (–6.6 to -0.9) | 0.010 |
| Diastolic blood pressure, mm Hg | 77.8 (1.0) | –2.5 (0.9) | 77.1 (1.0) | –1.9 (0.8) | –0.6 (–2.4 to 1.2) | 0.519 |
| Body mass index, kg/m$^2$ | 30.1 (0.5) | 0.12 (0.2) | 30.0 (0.5) | –0.04 (0.2) | 0.16 (–0.30 to 0.62) | 0.498 |
| Total cholesterol, mmol/L | 4.1 (0.1) | –0.08 (0.06) | 4.2 (0.1) | –0.15 (0.06) | 0.07 (–0.09 to 0.2) | 0.370 |
| HDL cholesterol, mmol/L | 1.25 (0.03) | –0.003 (0.018) | 1.26 (0.03) | 0.004 (0.018) | –0.007 (–0.054 to 0.039) | 0.754 |
| Completion of nine essential processes* | 65% (3.7) | –5.1% | 61% (3.8) | 3.4% | 0.78 (0.45 to 1.35) | 0.379 |
| HADS | 9.3 (0.5) | –1.05 (0.44) | 9.1 (0.5) | –0.60 (0.48) | –0.45 (–1.68 to 0.78) | 0.474 |
| DMSES† | 98.8 (2.4) | 2.93 (2.90) | 103.6 (2.3) | 1.38 (2.79) | 1.55 (–5.74 to 8.84) | 0.674 |
| DTSQ | 32.2 (0.6) | 0.94 (0.57) | 32.2 (0.6) | 0.45 (0.61) | 0.49 (–1.18 to 2.15) | 0.564 |

Results from multiply imputed data shown. Data are mean (SE) or mean difference (95% CI) unless otherwise specified.
*Percentage (SE) and OR (95% CI).
†Linear regression results shown due to lack of convergence for mixed model.
DMSES, Diabetes Management Self-Efficacy Scale; DTSQ, Diabetes Treatment Satisfaction Questionnaire; HADS, Hospital Anxiety and Depression Scale; HbA1c, glycated haemoglobin; HDL, high-density lipoprotein; HeLP-Diabetes, Healthy Living for People with Diabetes; PAID, Problem Areas in Diabetes.

appear to wane with time,[29] the effects of HeLP-Diabetes were greater at 12 months than at 3 months. There was no overall impact on diabetes-related distress, but some evidence that HeLP-Diabetes appeared to reduce distress in recently diagnosed individuals. However, it is worth noting that baseline PAID scores were exceptionally low in this trial population. In a small pilot study, participants offered supported access to HeLP-Diabetes reduced their PAID scores by six points (p=0.04) over 6 weeks.[30]

The trial has many strengths. It was a pragmatic trial, open to nearly all patients with T2DM in participating practices. Concealment of allocation was complete, as randomisation occurred after baseline data collection. Baseline prognostic factors were well balanced between groups. Every effort was made to achieve blinding, including requiring practices to have two nurses, so that data collection were undertaken by a nurse blind to participant allocation. Data for the coprimary outcomes at the primary outcome point were available for 78% and 86% of participants for HbA1c and PAID, respectively. All analyses were on an intention-to-treat basis, supplemented by a CACE analysis. Although response rates for the coprimary outcomes were good, some potential for bias existed. Our primary analysis used multiple imputation methods because evidence shows that the

assumptions underpinning this method are more defensible than those assumed using other approaches to missing data.[31] We also undertook sensitivity analyses including complete cases, non-contaminated cases and a linear model excluding centre; all yielded similar results.

The two coprimary outcomes reflected the twin aims of the intervention: to improve diabetes control and to reduce diabetes-related distress. Around 40% of patients with diabetes have significant levels of distress, which severely impacts on quality of life,[32] and diabetes-related distress is an important outcome for patients.[33] Our patient and public involvement panel were clear that this should be a primary outcome, and a recent meta-ethnography emphasised the importance of empowerment and quality of life in promoting long-term engagement with self-management.[34] In contrast, many healthcare professionals are more interested in glycaemic control. In line with previous trials in this area,[35] we decided to adopt both as coprimary outcomes and to test both at a 5% level of significance.[36]

There are some limitations. Despite maximising the inclusivity of the trial by minimising the exclusion criteria, participants were not representative of the overall population of patients with type 2 diabetes in England. Compared with the overall population, participants had

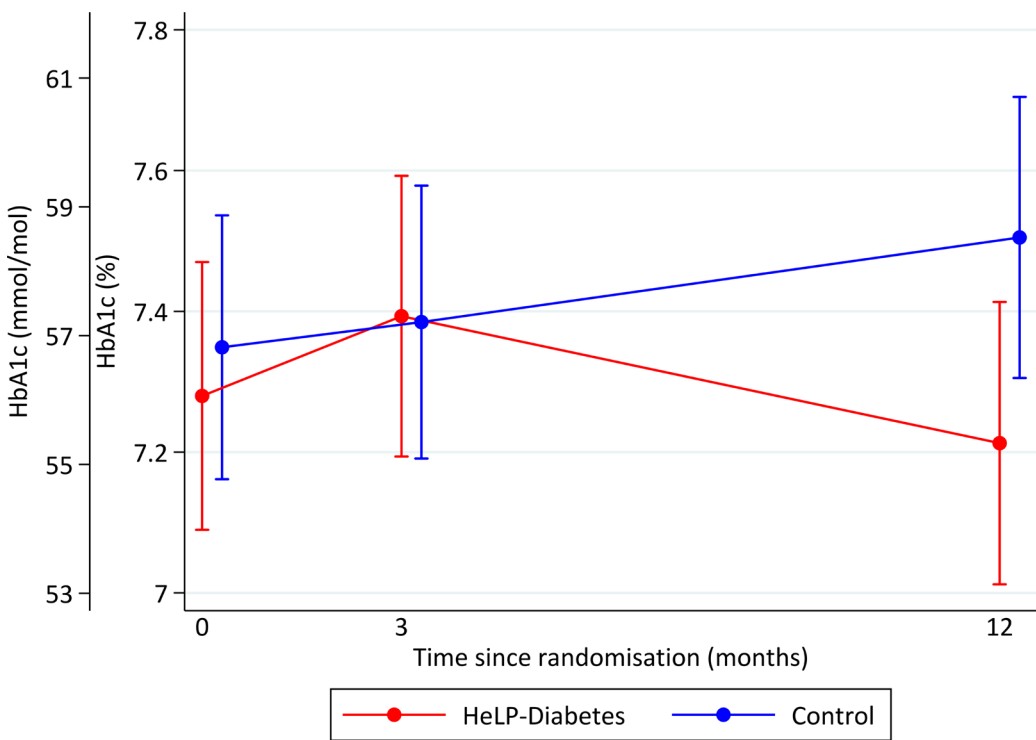

**Figure 2** Mean HbA1c (95% CI) over follow-up by randomised group using multiple imputation. HbA1c, glycated haemoglobin; HeLP-Diabetes, Healthy Living for People with Diabetes.

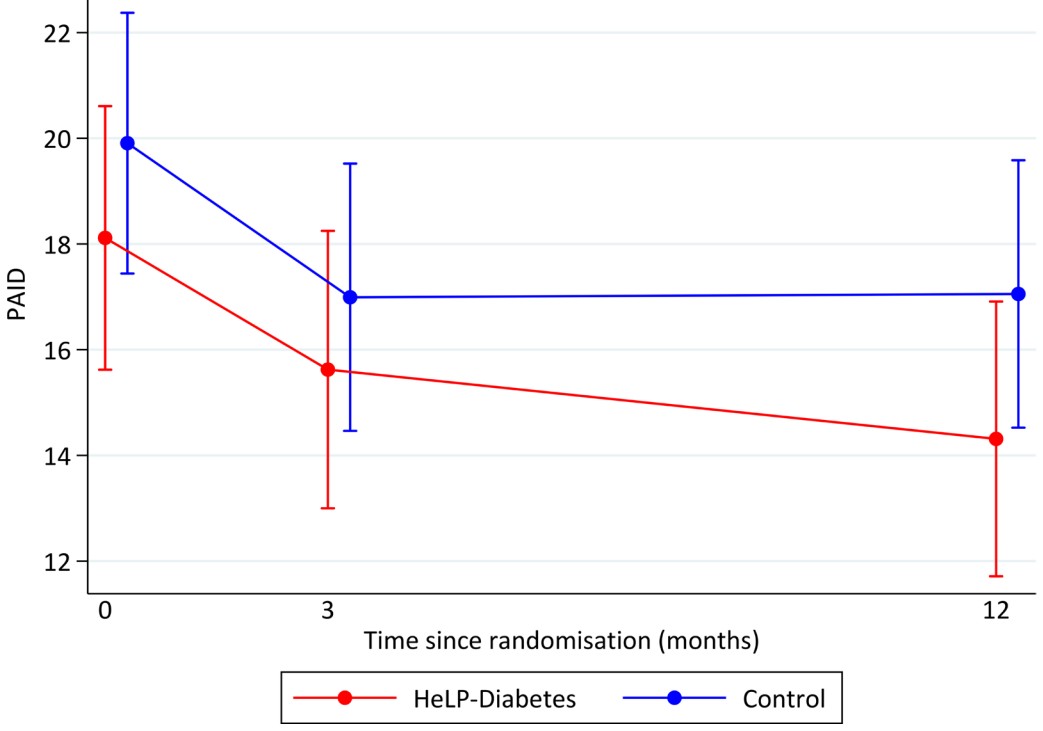

**Figure 3** Mean PAID score (95% CI) over follow-up by randomised group using multiple imputation. PAID, Problem Areas in Diabetes; HeLP-Diabetes, Healthy Living for People with Diabetes.

better control of their diabetes and cardiovascular risk factors,[6 7] and were much less distressed.[21] This finding mirrors that of a recent systematic review of demographic factors associated with web portal usage among people with diabetes which found that those with well controlled diabetes were more likely to use such portals than those with poor control.[37] However, fewer of our participants self-rated their computer skills as excellent (57% of our sample compared with a national average of 73%).[12] This good control at baseline has two implications—first, that

**Table 3** Extent of website usage over 12-month follow-up

| | HeLP-Diabetes | Control | p Value* | N missing |
|---|---|---|---|---|
| No of log-ins per person | 18.7 (84.0) | 4.8 (8.0) | 0.0001 | 0 |
| No pages visited per log-in | 10.5 (6.7) | 7.7 (5.0) | <0.0001 | 105† |
| Time spent in each log-in (min)‡ | 12.3 (9.8) | 8.2 (8.4) | <0.0001 | 105† |
| No of days in which website was accessed over follow-up | 10.1 (22.9) | 3.3 (5.1) | <0.0001 | 0 |

Mean (SD) unless otherwise specified.
*Wilcoxon rank-sum test.
†105 individuals did not log in after their facilitation visit (42 intervention, 63 control).
‡Measured as time from first page accessed to last page accessed within a log-in session.
HeLP-Diabetes, Healthy Living for People with Diabetes.

there was little room for improvement in this population and second, that this population may have been unusually motivated to self-manage their diabetes. Although every effort was made to maintain blinding, it is possible that some participants may have discussed their use of the intervention with research nurses, making it possible to infer which arm they had been allocated to. This could have affected research nurses' measurements of secondary clinical outcomes, such as blood pressure or weight, but could not have affected assessment of HbA1c as this was measured by laboratory staff who were blinded. There appeared to be high potential for contamination between two participants who shared the same surname and address, and a further two participants did not receive their allocated intervention due to an error at practice level; excluding these four made no difference to the results. A further limitation of the trial is that it provides little insight into the mechanism of action of HeLP-Diabetes. This was the result of a deliberate decision to focus on clinically important outcomes and minimise both the response burden and the potential impact of measurement on participants.

This is the first UK-based trial of a web-based self-management programme for people with type 2 diabetes, and internationally, the first trial of such a comprehensive intervention that aims to address the three main tasks of self-management: emotional, medical and role management.[19] In the Cochrane review of computer-based self-management interventions for people with T2DM, only four of the included studies had follow-up of 12 months or more.[14] Of these, three interventions were clinic-based, with participants completing self-assessment tools on a touch screen and receiving tailored advice during their baseline visit to their diabetes clinician[38–40]

and one was a mobile phone-based intervention which provided tailored messages in response to participant's results of blood glucose self-monitoring data.[41] A more recent systematic review of internet delivered diabetes self-management identified 2 trials with 12 or more months follow-up.[42] One trial was on a structured intervention based on a peer-led, group-based, diabetes self-management course.[43] There were six sessions, with each session available for 1 week. Each session required participants to make a specific action plan to address a problem they were experiencing. Peer facilitators encouraged use of the programme. Follow-up was planned at 6 and 12 months; however, HbA1c data were only available at 6 months. The other trial compared two versions of a web-based intervention (with and without additional social support) to enhanced usual care. The web-based intervention was designed using social cognitive theory and a social ecological model, with a focus on three main behaviours: dietary intake, physical activity and medication adherence. Users of either web-based intervention received motivational phone calls to encourage adherence and development of action plans. Those randomised to the enhanced intervention (with additional social support) received two additional phone calls and an invitation to attend a group session. There was no difference between groups in HbA1c or other biological outcomes at 12 months.[44] Thus, the results of this trial add significantly to the available literature.

On the basis of these results, HeLP-Diabetes may be considered as an addition to the current menu of self-management support for people with type 2 diabetes, and may help increase overall access and uptake. Most commissioned services currently focus on newly diagnosed patients, leaving clear unmet need for people who have had their diabetes for longer, but are looking for ways to improve their health. Many patients are not ready to engage in self-management early in their illness journey,[9] but become motivated to do so later, often as a result of a change in medication or development of a complication.[45] The intervention is low cost, and as most costs are fixed, irrespective of number of users, is likely to be cost-effective, particularly if widely used. A cost-effectiveness analysis of HeLP-Diabetes will be reported separately.

**Author affiliations**
[1]Research Department of Primary Care and Population Health, University College London, London, UK
[2]Department of Public Health and Primary Care, Cardiovascular Epidemiology Unit, University of Cambridge, Cambridge, UK
[3]Department of Psychology, University of Bath, Bath, UK
[4]Population Health Research Institute, St George's, University of London, London, UK
[5]Department of Health Sciences, University of York, York, UK
[6]Whittington Health, London, UK
[7]Nuffield Department of Primary Care Health Sciences, University of Oxford, Oxford, UK
[8]Department of Clinical, Educational and Health Psychology, Centre for Behaviour Change, University College London, London, UK
[9]Department of Psychology, University of Southampton, Southampton, UK
[10]Faculty of Health Sciences, University of Southampton, Southampton, UK

**Acknowledgements** We gratefully acknowledge the permission to use under license the validated behaviour change modules for weight loss (POWeR), alcohol reduction (DownYour Drink) and smoking cessation (StopAdvisor) and the diabetes module from Healthtalk online in HeLP-Diabetes. We are grateful to Orla O'Donnell for outstanding project management, Fiona Giles for administrative support, all our PPI who contributed to the development, maintenance and delivery of the intervention and/or the management and oversight of the trial, staff at the participating practices, Primary Care Research Network (PCRN) staff and all our participants.

**Contributors** EM, MS, CD, KP, MB, AF, SM, LY, CM, SP, FS and DP all contributed to the design of the trial. MS, assisted by MH and KM, designed the statistical analysis plan and undertook the analysis. SP and JL designed and undertook the health economic aspects of the trial. EM, KP, CD, JR, GA, LY and SM contributed to the development and delivery of the intervention. MK contributed PPI input. EM and MS wrote the first draft of the paper; all authors commented on this draft and approved the final version.

**Funding** This paper presents independent research funded by the National Institute for Health Research (NIHR) under its Programme Grants for Applied Research Programme (grant reference number RP-PG-0609-10135). Work conducted at the Cardiovascular Epidemiology Unit, University of Cambridge by MS and MH was additionally funded by the UK Medical Research Council (MR/L003120/1), British Heart Foundation (RG/13/13/30194) and UK National Institute for Health Research Cambridge Biomedical Research Centre. AF is an NIHR Senior Investigator and receives funding from Oxford NIHR Biomedical Research Centre.

**Disclaimer** The views expressed are those of the author(s)and not necessarily those of the NHS, the NIHR or the Department of Health. The funder had no role in the study design, data collection, data analysis, data interpretation or writing of the report. All authors had full access to all the data in the study and can take responsibility for the integrity of the data and data analysis. The lead author had final responsibility for the decision to submit for publication, affirms that the manuscript is an honest, accurate and transparent account of the study being reported; that no important aspects of the study have been omitted; and that there were no significant discrepancies from the published protocol.

**Competing interests** EM is the managing director of a not-for-profit community interest company established to disseminate HeLP-Diabetes across the NHS.

**Patient consent** Detail has been removed from this case description/these case descriptions to ensure anonymity. The editors and reviewers have seen the detailed information available and are satisfied that the information backs up the case the authors are making.

**Ethics approval** Camden and Islington National Research Ethics Service Committee.

**Provenance and peer review** Not commissioned; externally peer reviewed.

**Data sharing statement** Patient-level data, the full dataset and statistical code are available from the corresponding author. Consent for data sharing was not obtained from participants, but the potential benefits of sharing these data outweigh the potential harms as the data are anonymised.

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
