## [Reviewer comments · BMJ Open]

ARTICLE DETAILS

TITLE (PROVISIONAL)	Web-based self-management support for people with type 2 diabetes (HeLP-Diabetes): randomised controlled trial in English primary care.
AUTHORS	Murray, Elizabeth; Sweeting, Michael; Dack, Charlotte; Pal, Kingshuk; Modrow, Kerstin; Hudda, Mohammed; Li, Jinshuo; Ross, Jamie; Alkhaldi, Ghadah; Barnard, Maria; Farmer, Andrew; Michie, Susan; Yardley, Lucy; May, Carl; Parrott, Steve; Stevenson, Fiona; Knox, Malcolm; Patterson, David

VERSION 1 - REVIEW

REVIEWER	Norbert Hermanns Research Institute Diabetes (FIDAM) Bad Mergentheim & University of Bamberg Germany
REVIEW RETURNED	31-Jan-2017

GENERAL COMMENTS	This randomised trial tested the impact of a web-based self-management programme (HeLP-Diabetes) in comparison to a control group, which had access to a simple text-based website about diabetes. From 374 randomised participants, HbA1c follow-up data was available from 294 subjects and PAID follow-up data was available from 321 participants within the prespecified time window of 10 -14 months. There was a significant HbA1c difference by 0.24 percentage points and a non-significant difference by -1.5 points regarding the PAID scores. The study meets high methodological standards and the article is well written. There are some points which could improve the manuscript further. 1) A sceptic person might argue that the study effect regarding HbA1c is mainly due to the deterioration of participants from the control group. May the authors should discuss possible reasons for the observed deterioration by 0.16 percentage points.2) How do the authors explain the initial deterioration of HbA1c values in the first 3 months?3) Do the authors have any mechanistic idea which factors could be responsible for the observed difference in HbA1c values?4) Could the authors provide an unadjusted analysis for the impact of the interventions on HbA1c and PAID scores in the supplementary section?5) Do the authors have any information about diabetes treatment at baseline and treatment changes at follow-up? If not, the authors should mention this as a limitation.6) Given the small improvement of glycaemic control (by 0.08 percentage points), which might be due to the positive selection of
---

	the sample, the HeLP programme rather seems to support the maintenance of good glycaemic control in already motivated people with diabetes than improve glycaemic control in poorly controlled people with diabetes. Can the authors comment on this? In summary, this manuscript provides interesting new information. The article is well written and the points raised can be easily addressed by the authors.
--	---

REVIEWER	Catherine Yu St. Michael's Hospital, Canada
REVIEW RETURNED	01-Feb-2017

GENERAL COMMENTS	Summary: This is a well-written, easy-to-read manuscript on a very clinically relevant topic. In addition, the study is well-designed with patient input starting at conception, and well-reported, using previously published protocol, and following CONSORT and TIDIER reporting guidelines. I cannot comment on statistical analyses. Correction of minor formatting issues and modification of interpretation and tone of findings would improve the paper. General: - Minor formatting issues (e.g. first line of Methods, spacing between paragraphs, misspelling of facilitation page 8, capitalization of titles e.g. Appendix 1, definition of abbreviations e.g. ccg) Abstract - The first two points of “Strengths and Limitations” are not actually strengths and limitations of current study, though authors have outlined these clearly in their discussion. Methods: - The selected effect size is small which authors acknowledge; however, see interpretation under “Results” and “Discussion” below. - I defer review of statistical analyses. - Results: - The study had an excellent follow-up rate. Is this consistent with that in the literature? This should be discussed in the Discussion - Although statistically significant, mean difference found is less than the minimally clinically important difference defined by authors in sample size calculation. This should be discussed in the Discussion Discussion: - As described above, discussion should include excellent follow-up rate, as well as clinical significance of primary outcome - Could consider re-ordering Discussion by moving the 5th paragraph, before strengths/limitations - Final paragraph: given limitations noted by authors (generalizability, limited usage, mean difference less than minimal clinically important difference), first sentence of concluding paragraph perhaps should be softened with: “may be considered”, as well as a phrase regarding optimization of usage.
--

REVIEWER	Frank Snoek VU University Medical Centre and Academic Medical Centre, Amsterdam, The Netherlands
REVIEW RETURNED	01-Feb-2017

GENERAL COMMENTS	General comment: A well-designed timely study, sufficiently powered, and conducted in a real-world setting (20 UK primary care practices) with 12 mos. follow up. Inclusion (n=374 included out of 421=89%) took 15 months mos. Co-primary outcomes: HbA1c and diabetes-distress. The efficacy of web-based self-management program (HeLP-Diabetes) was compared to an information website, on top of care as usual. The findings at 12 mos. Demonstrate difference in glycaemic control in favour of the intervention. No difference in diabetes distress (PAID score) or any of the secondary outcomes. The difference in HbA1c was mainly due to worsening of glycaemic control in the control condition, particularly after 3 mos. Baseline HbA1c was relatively good (7.26/7.35%) with little room for improvement. Likewise, diabetes-distress levels were low, with mean score of 19 (% above cut-off of 40 on PAID scale is not revealed). The limitations and strengths of this study are well-discussed. Specific comments:  - Measures. Given the aim of the programme (enhance self-management) it is surprising to see no measure of self-care was included (other than self-efficacy). We may assume that improvement in glycaemic control is mediated by improved self-care. It now remains uncertain why the control group did (somewhat) worse; - DTSQ: what treatment is being evaluated ? please explain why this measure was used; - Target group. The program was offered to primary care (both newly diagnosed and those with longer disease duration) DM2 patients, not necessarily in need of additional support. Previous self-management education was also not an exclusion criterion. The authors acknowledge this limitation, but I have difficulty understanding why not those in poor control and/or distressed were approached, as they are the ones that can profit from this type of support; More so if cost are taken into account; - Prevalence of distress. The authors took diabetes-distress as a co-primary, as suggested by the patient panel, but seriously overestimated the scope of the problem in primary care, assuming 40% highly distressed -referring to the DAWN2 survey (2013). There is evidence to suggest that diabetes-distress in primary care is significantly lower than in secondary care e.g. in The Netherlands 4% reported high distress (Stoop et al., 2014). - Internet. Interestingly, not having internet access at home was not a an exclusion criterion, that is not further explained. Were they supposed to use internet cafes, open wifi areas? - Treatment. Information on the treatment regimen is lacking. Please add. One alternative (although maybe not likely) explanation for better HbA1c in intervention group relative to controls could be that they were better/more intensively treated, possibly as an effect of more consultations/contact due to patient engagement – also see comment above on self-care: perhaps medication adherence improved? - The web-based program appears well-grounded in theory and offers a number of modules aimed to strengthen self-management of the participants in the domains of self-regulation concerning the
---

	disease, emotions and role functioning. It does strike me as a patchwork of existing content and functionalities , referred to as “evidence-based” – while there is proof of effectiveness for certain self-management programs, the evidence for individual modules within these programs on their own, I think is weak or non-existent as these modules have never been tested as such. - Usage. The user-data are interesting, but it would be helpful to see data on usage of the most/least popular modules and how these were used over time. It is to be expected that certain modules are used at different times and differently over time. Such data can help to further improve and tailor the intervention to the persons’ needs. - Suggestions. The authors make little effort to make suggestions for further research and improvement of the program. Is there nothing to suggest? I would recommend to test whether some sections can be omitted or improved (e.g. by using EMA, apps), and certainly this programme needs to be tested in a more mixed, problematic patient group before recommending implementation.
--	---

REVIEWER	Javier Mariani Hospital El Cruce. Florencio Varela, Buenos Aires, Argentina.
REVIEW RETURNED	15-Feb-2017

GENERAL COMMENTS	Authors report a randomized controlled trial assessing the effects a web-based self-management. Main analysis comparing HbA1c and PAID scale were conducted using linear effects model with multiple imputation of missing values. Overall, statistical methods are sound, however it would be informative to present (as sensitivity analyses) a simpler comparison between groups for a randomized controlled trial with balanced distribution of potential confounders. In this sense, i suggest to add a t test or Mann-Whitney’s U test (as appropriate) for the primary end point, without imputation methods to allow a more direct interpretation of the results to non-expert reader.
--

REVIEWER	Chris Penfold University of Bristol, UK
REVIEW RETURNED	10-Apr-2017

GENERAL COMMENTS	This is a well written paper describing results from a web-based RCT of self-management support for people with type 2 diabetes. The authors have followed their pre-registered protocol and have generally reported the results appropriately. My main comments relate to the CONSORT diagram. Main comments: - Figure 1 CONSORT diagram: 47 people were withdrawn prior to randomisation since they did not complete baseline questionnaires. Although in the protocol it was stated that randomisation would be undertaken after baseline data collection, I am not clear why completing baseline data collection was a requirement for people to be randomised. Could the authors please clarify the reasons for the exclusion of these people. - page 9, line 55: Dependent on the response to the above point, the authors may need to include a further limitation regarding potential
---

	selection bias as a result of excluding people who consented but did not complete the baseline questionnaire. - Figure 1 CONSORT diagram (layout): Could the authors please give reasons why people withdrew or were withdrawn from the trial before 60 days, or 60-304 days. Please remove the footnote 'Withdrawn from the trial after 304 days', which does not seem to have been used. It would be helpful to include the total number missing data at each timepoint where there are multiple reasons (e.g. total number missing HbA1c or 'Training date' at baseline). The '12 month nurse visit' boxes are not used in the text and should be removed from this figure. It would be helpful if the results collected in the required timeframes were more prominent - i.e. intervention arm 12-month HbA1c - 'n=144' should be the prominent figure, and n=11 should be included below this in brackets with suitable footnotes. This applies to the '12 month PAID' and '3 month nurse visit' boxes too. Minor comments: - page 8, line 41: 'benoted' should be 'be noted' - page 10, line 12: Please revise this line to clarify that it is the research nurse's measurement of these secondary outcomes which could have been affected, not the outcomes themselves. Possible wording 'This could have affected the research nurse's measurement of secondary clinical outcomes, such as...'. Also, please expand on why assessment of glycated haemoglobin could not have been affected by failure of blinding. - page 10, line 45: Please remove the phrase 'statistically significant'
--	--

REVIEWER	Resmi Gupta Cincinnati Children's Hospital Medical Center USA
REVIEW RETURNED	11-Apr-2017

GENERAL COMMENTS	The purpose of the study was to determine the effectiveness of a web-based self-management program for people with type 2 diabetes in reducing diabetes related distress and improving glycemic control. As a Biostatistician, I will concentrate my review on the Analysis and Results portion of the paper. While the study has strengths, it also has few weaknesses. Below are my comments. (1) It is not clear what kind of statistical model was used for the inference. I understand that it was linear mixed effects model with center as random effect, but were any nested effects tested? (for example: patients nested within center). How many random effects were in the model? Please provide sufficient details. (2) What kind of covariance structure was used? The authors are advised to mention the covariance structure and the reason for choosing so. (3) For GEE, what was the assumed correlation structure? Were robust (sandwich) variance estimators used? Please provide sufficient details. (4) It seems that time by group comparison was conducted; but I didn't understand what the number of tests was, or the
---

	corresponding significance threshold, especially when the authors appear to use 0.05 as their significance level throughout the manuscript. (5) Was normality assumption checked for the continuous outcome variables? If not, the authors are advised to check the model assumption using histogram/ or qqplot before choosing the model. (6) The text ... "secondary outcome measures were analyzed similarly using generalized linear mixed model " ... this sentence is not clear. What the distribution looks like for secondary outcomes? Please provide clear details. (7) Do the figures (2, 3) present marginally adjusted means from the regression models? If so, this should be described in the methods section.
--	---

VERSION 1 – AUTHOR RESPONSE

Reviewer comment	Response
Reviewer: 1 Reviewer Name: Norbert Hermanns	
The study meets high methodological standards and the article is well written. There are some points which could improve the manuscript further.	Thank you.
A sceptic person might argue that the study effect regarding HbA1c is mainly due to the deterioration of participants from the control group. May the authors should discuss possible reasons for the observed deterioration by 0.16 percentage points.	Yes, but this deterioration is the normal clinical trajectory for people with type 2 diabetes. Avoiding this deterioration is a real achievement.
How do the authors explain the initial deterioration of HbA1c values in the first 3 months?	As above, deterioration is to be expected in a population of people with type 2 diabetes. It takes time to change patterns of thoughts, feelings and behaviours, and in turn, it takes time for these changes to impact on HbA1c.
Do the authors have any mechanistic idea which factors could be responsible for the observed difference in HbA1c values?	This trial was designed to measure effectiveness, and we deliberately did not include proximal outcomes along the proposed pathway of action, in order to keep the response burden on participants to a minimum.
Could the authors provide an unadjusted analysis for the impact of the interventions on HbA1c and PAID scores in the supplementary section?	These results have now been added as a further sensitivity analysis (Sensitivity Analysis 6) to Supplementary Table 3. The magnitude of the effects are comparable to previous analyses, though the confidence intervals are wider due to the unadjusted analyses not accounting for

	baseline measures.
Do the authors have any information about diabetes treatment at baseline and treatment changes at follow-up? If not, the authors should mention this as a limitation.	We have information about medication and service use which was used for the health economic analysis. It was not used for the effectiveness analysis. Both groups received treatment as usual from their clinicians. It is possible that the intervention led participants to either request more treatment, or to adhere better to treatment – as described above we did not design the trial to explore mechanisms of action. This is mentioned as a limitation on page 10.
Given the small improvement of glycaemic control (by 0.08 percentage points), which might be due to the positive selection of the sample, the HeLP programme rather seems to support the maintenance of good glycaemic control in already motivated people with diabetes than improve glycaemic control in poorly controlled people with diabetes. Can the authors comment on this?	The subgroup analysis of patients with an HbA1c of 7.5% or more showed a benefit (reduction) of -0.51% (-0.93 to -0.09), (Supplementary Table 4) suggesting that the intervention may have more impact in those who are less well controlled. However as the p-value for interaction was not significant, we have been careful not to over claim or over interpret these results.
In summary, this manuscript provides interesting new information. The article is well written and the points raised can be easily addressed by the authors.	Thank you.
Reviewer: 2 Reviewer Name: Catherine Yu	
This is a well-written, easy-to-read manuscript on a very clinically relevant topic. In addition, the study is well-designed with patient input starting at conception, and well-reported, using previously published protocol, and following CONSORT and TIDIER reporting guidelines. I cannot comment on statistical analyses. Correction of minor formatting issues and modification of interpretation and tone of findings would improve the paper.	Thank you.
General: Minor formatting issues (e.g. first line of Methods, spacing between paragraphs, misspelling of facilitation page 8, capitalization of titles e.g. Appendix 1, definition of abbreviations e.g. ccg)	Thank you, we have tried to address these. Facilitation has been corrected on page 8 and the definition of CCG provided in Appendix 1.

Abstract: The first two points of “Strengths and Limitations” are not actually strengths and limitations of current study, though authors have outlined these clearly in their discussion.	This has been changed.
Methods: The selected effect size is small which authors acknowledge; however, see interpretation under “Results” and “Discussion” below. I defer review of statistical analyses.	See below.
Results: The study had an excellent follow-up rate. Is this consistent with that in the literature? This should be discussed in the Discussion Although statistically significant, mean difference found is less than the minimally clinically important difference defined by authors in sample size calculation. This should be discussed in the Discussion	We are not quite sure what the reviewer means by this – follow-up rates vary widely between trials reported in the literature. We certainly put every effort into ensuring good follow-up and are pleased by our success. The clinical significance of the change in HbA1c is discussed in the Discussion (paragraph 1).
Discussion:  - As described above, discussion should include excellent follow-up rate, as well as clinical significance of primary outcome - Could consider re-ordering Discussion by moving the 5th paragraph, before strengths/limitations - Final paragraph: given limitations noted by authors (generalizability, limited usage, mean difference less than minimal clinically important difference), first sentence of concluding paragraph perhaps should be softened with: “may be considered”, as well as a phrase regarding optimization of usage. 	The discussion does mention the follow-up rates (bottom of page 9). We have followed standard BMJ advice on ordering of the discussion. If the editors wish us to change the order of the discussion we are happy to do so. We have made this suggested change.
Reviewer: 3 Reviewer Name: Frank Snoek	
General comment: A well-designed timely study, sufficiently powered, and conducted in a real-world setting (20 UK primary care practices) with 12 mos. follow up. Inclusion (n=374 included out of 421=89%) took 15 months. Co-primary outcomes: HbA1c and diabetes-distress. The efficacy of web-based self-management program (HeLP-Diabetes) was compared to an information website, on top of care as usual. The findings at 12 mos. Demonstrate difference in glycaemic control in favour of the intervention. No difference in diabetes distress (PAID score)	Thank you.

or any of the secondary outcomes. The difference in HbA1c was mainly due to worsening of glycaemic control in the control condition, particularly after 3 mos. Baseline HbA1c was relatively good (7.26/7.35%) with little room for improvement. Likewise, diabetes-distress levels were low, with mean score of 19 (% above cut-off of 40 on PAID scale is not revealed). The limitations and strengths of this study are well-discussed.	
Specific comments: Measures. Given the aim of the programme (enhance self-management) it is surprising to see no measure of self-care was included (other than self-efficacy). We may assume that improvement in glycaemic control is mediated by improved self-care. It now remains uncertain why the control group did (somewhat) worse;	Yes. As described above, the decision not to include proximal outcomes along the pathway of action was deliberate, in order to minimise the response burden on participants. The decline in HbA1c in the comparator group is to be expected clinically.
DTSQ: what treatment is being evaluated? please explain why this measure was used;	This is a measure of satisfaction with overall treatment. The instructions for completing this questionnaire state: "By diabetes treatment we mean the whole package of care you get from you general practice, hospital clinic or community clinic, including the medicines and any help you get with learning to look after your diabetes". We used this questionnaire as an outcome, to explore whether the intervention impacted on satisfaction with overall treatment.
Target group. The program was offered to primary care (both newly diagnosed and those with longer disease duration) DM2 patients, not necessarily in need of additional support. Previous self-management education was also not an exclusion criterion. The authors acknowledge this limitation, but I have difficulty understanding why not those in poor control and/or distressed were approached, as they are the ones that can profit from this type of support; More so if cost are taken into account;	We designed the trial to be pragmatic, and to mirror, as closely as possible, how we would expect this intervention to be implemented, if it were to be widely commissioned. Normal practice is to offer such interventions to everyone, irrespective of baseline control. We did approach those with poor control and / or distress; however we could only recruit those who actively consented to participate in the trial.
Prevalence of distress. The authors took diabetes-distress as a co-primary, as suggested by the patient panel, but seriously overestimated the scope of the problem in primary care, assuming 40% highly distressed -referring to the DAWN2 survey (2013). There is evidence to suggest that diabetes-distress in primary care is significantly lower than in secondary care e.g. in The Netherlands 4% reported high distress (Stoop et al., 2014).	Thank you – this is interesting. The DAWN2 report states: "recruitment ensures representation of the diabetes population in terms of geographical distribution, age, gender, education and disease status". We are not aware of UK data suggesting a

	difference in levels of distress between patients treated in primary and secondary care, and as the great majority of UK patients with T2DM are treated in primary care, we believed the DAWN2 data to be the best available for comparison.
Interestingly, not having internet access at home was not an exclusion criterion, that is not further explained. Were they supposed to use internet cafes, open wifi areas?	Yes. This was part of making the trial pragmatic and aiming for wide inclusion criteria, which mirrored likely scenarios for subsequent implementation. The UK is well provided with free public internet access, e.g. through libraries.
Treatment. Information on the treatment regimen is lacking. Please add. One alternative (although maybe not likely) explanation for better HbA1c in intervention group relative to controls could be that they were better/more intensively treated, possibly as an effect of more consultations/contact due to patient engagement – also see comment above on self-care: perhaps medication adherence improved?	As discussed above, we deliberately did not collect data on proximal outcomes along the proposed pathway of action. We agree that the intervention could have worked by encouraging participants in the intervention group to be more adherent to medication (a targeted behaviour), and / or more willing to increase medication / intensify treatment. Both these would be manifestations of improved self-management.
The web-based program appears well-grounded in theory and offers a number of modules aimed to strengthen self-management of the participants in the domains of self-regulation concerning the disease, emotions and role functioning. It does strike me as a patchwork of existing content and functionalities, referred to as “evidence-based” – while there is proof of effectiveness for certain self-management programs, the evidence for individual modules within these programs on their own, I think is weak or non-existent as these modules have never been tested as such.	We use the term “evidence-based” as we used best available evidence to underpin all the decisions made during the development process, e.g. evidence on good practice in development of digital interventions, evidence to support engagement, etc.
Usage. The user-data are interesting, but it would be helpful to see data on usage of the most/least popular modules and how these were used over time. It is to be expected that certain modules are used at different times and differently over time. Such data can help to further improve and tailor the intervention to the persons’ needs.	Yes, but this is not the focus of this paper.
Suggestions. The authors make little effort to make suggestions for further research and improvement of the program. Is there nothing to suggest? I would recommend to test whether some sections can be omitted or improved (e.g. by using EMA, apps), and certainly this	This paper reports one study of a large programme of research. The programme of research is reported in a monograph, currently under editorial review at NIHR.

programme needs to be tested in a more mixed, problematic patient group before recommending implementation.	
Reviewer: 4 Reviewer Name: Javier Mariani	
Authors report a randomized controlled trial assessing the effects a web-based self-management. Main analysis comparing HbA1c and PAID scale were conducted using linear effects model with multiple imputation of missing values. Overall, statistical methods are sound, however it would be informative to present (as sensitivity analyses) a simpler comparison between groups for a randomized controlled trial with balanced distribution of potential confounders. In this sense, i suggest to add a t test or Mann-Whitney's U test (as appropriate) for the primary end point, without imputation methods to allow a more direct interpretation of the results to non-expert reader.	A number of simpler analyses are presented in Supplementary Table 3 for comparison of the randomised groups. As requested by another reviewer, we have also added an unadjusted analysis (Sensitivity Analysis 6), in which a linear regression is performed only accounting for randomised group. This gives the same p-value as a simple t-test between groups. Note, however, that power is lost in these analyses as we do not account for baseline measures of the outcome.
Reviewer: 5 Reviewer Name: Chris Penfold This is a well written paper describing results from a web-based RCT of self-management support for people with type 2 diabetes. The authors have followed their pre-registered protocol and have generally reported the results appropriately. My main comments relate to the CONSORT diagram.	Thank you.
Main comments: - Figure 1 CONSORT diagram: 47 people were withdrawn prior to randomisation since they did not complete baseline questionnaires. Although in the protocol it was stated that randomisation would be undertaken after baseline data collection, I am not clear why completing baseline data collection was a requirement for people to be randomised. Could the authors please clarify the reasons for the exclusion of these people.	The CONSORT diagram has been updated (see below). We required participants to complete baseline data prior to randomisation for two reasons: the first was to ensure complete concealment of allocation and prevent bias in completion of baseline questionnaires. The second was to ensure that recruited participants were sufficiently engaged with the trial to complete questionnaires.
page 9, line 55: Dependent on the response to the above point, the authors may need to include a further limitation regarding potential selection bias as a result of excluding people who consented but did not complete the baseline questionnaire.	As with all RCTs, the results are interpretable to the population that are randomised (they are unbiased for this population). Whether the results generalise to the wider T2D population is for open discussion. As mentioned above, one reason for randomising after collecting baseline data is to ensure we have an engaged cohort of

	patients who would be more likely to return at 3 and 12 months, and also to avoid having too much missing baseline data.
Figure 1 CONSORT diagram (layout): Could the authors please give reasons why people withdrew or were withdrawn from the trial before 60 days, or 60-304 days. Please remove the footnote 'Withdrawn from the trial after 304 days', which does not seem to have been used.	The reasons for withdrawal have been added as a footnote to Figure 1, and the 'Withdrawn from the trial after 304 days' removed.
It would be helpful to include the total number missing data at each timepoint where there are multiple reasons (e.g. total number missing HbA1c or 'Training date' at baseline). The '12 month nurse visit' boxes are not used in the text and should be removed from this figure.	The total number missing HbA1c or Training date at baseline and 3-months are given in Figure 1. The total number with completed outcomes at 12-months are also given at the bottom of Figure 1.
It would be helpful if the results collected in the required timeframes were more prominent - i.e. intervention arm 12-month HbA1c - 'n=144' should be the prominent figure, and n=11 should be included below this in brackets with suitable footnotes. This applies to the '12 month PAID' and '3 month nurse visit' boxes too.	As requested we have swapped round the numbers, so that those that attended in the 10-14 month timeframe are the main results presented and those that attended anytime beyond 10 months are presented in parentheses. To aid legibility, we have accepted the changes in Figure 1, so the changes are not immediately visible.
Minor comments: - page 8, line 41: 'benoted' should be 'be noted' - page 10, line 12: Please revise this line to clarify that it is the research nurse's measurement of these secondary outcomes which could have been affected, not the outcomes themselves. Possible wording 'This could have affected the research nurse's measurement of secondary clinical outcomes, such as...'. Also, please expand on why assessment of glycated haemoglobin could not have been affected by failure of blinding. - page 10, line 45: Please remove the phrase 'statistically significant'	Thank you, this has been corrected. This has been done. This has been done.

	This has been done.
Reviewer: 6 Reviewer Name: Resmi Gupta	
(1) It is not clear what kind of statistical model was used for the inference. I understand that it was linear mixed effects model with center as random effect , but was any nested effects tested ? (for example: patients nested within center) . How many random effects were in the model ? Please provide sufficient details .	The linear mixed model had a random centre term (with 21 random-effects corresponding to the 21 centres). A sensitivity analysis excluding these random-effects has been conducted (Table S3; sensitivity analysis 5) and results were almost identical to the linear mixed model suggesting little between-centre variability. However, our protocol pre-specified a linear mixed model so this was retained for the primary analysis. The text stating there were 20 centres has been corrected (this was an error, and we are grateful for the opportunity to pick this up).
(2) What kind of covariance structure was used ? The authors are advised to mention the covariance structure and the reason for choosing so.	Only one random-effect term (a random centre effect) was included in the model and so there was no covariance structure between random-effects. Specifically $y_{ik} = \beta_0 + \beta^T X_{ik} + b_k + \epsilon_{ik}$ for patient i in centre k and where b_k are random centre-effects and the β are fixed effects (including fixed effect for randomised arm).
(3) For GEE, what was the assumed correlation structure? Were robust (sandwich) variance estimators used? Please provide sufficient details.	No GEE estimation was performed.
(4) It seems that time by group comparison was conducted ; but I didn't understand what the number of tests was, or the corresponding significance threshold, especially when the authors appear to use 0.05 is their significance level throughout the manuscript.	The two primary outcome measures were HbA1c and PAID at 12-months (both regressed against baseline values and other baseline covariates), and both were tested at a significance threshold of 0.05. These are the two main significance tests that were pre-specified. Three-month outcome data and other secondary

	outcomes are presented together with their p-values for a comparison of groups; however no strict significance thresholds were applied to these secondary outcomes and interpretation of these results are purely hypothesis generating. The text reporting secondary outcomes has been altered to make this clearer. (page 8).
(5) Was normality assumption checked for the continuous outcome variables? If not, the authors are advised to check the model assumption using histogram/ or qqplot before choosing the model.	Normality was assessed for both outcomes. For HbA1c, there was some evidence of non-normality from a histogram and qqplot and an inverse power of HbA1c gave the best transformation to a normally distributed variable. However, regressing this transformed outcome against the equivalent transformed baseline and randomised group gave a near-equivalent p-value for the effect of randomised group. Therefore due to interpretability, it was decided to present regression coefficients from the untransformed regression model.
(6) The text ... "secondary outcome measures were analyzed similarly using generalized linear mixed model" ... this sentence is not clear. What the distribution looks like for secondary outcomes? Please provide clear details.	We have added more detail to the methods section (page 7)
(7) Do the figures (2, 3) present marginally adjusted means from the regression models? If so, this should be described in the methods section.	Figures 2 and 3 show the empirical means and confidence intervals (assuming normality) of the outcomes at each time point, stratified by randomised group, and estimated over multiply imputed datasets. Therefore they are not based on marginal means from the multivariate regression models described above.

VERSION 2 – REVIEW

REVIEWER	Norbert Hermanns Research Institute of the Diabetes Academy Mergentheim (FIDAM)
REVIEW RETURNED	22-May-2017

GENERAL COMMENTS	The authors did a nice Revision, addressing all points raised in my review sufficiently.
--

REVIEWER	Yu, Catherine St. Michael's Hospital, Canada
REVIEW RETURNED	30-May-2017

GENERAL COMMENTS	The authors have addressed the majority of issues raised by the prior review. One outstanding concern is the authors' interpretation of the clinical significance of their findings. They state that a minimal significant change in A1c would be 0.25% in their sample size calculation, yet achieve a smaller change: a mean between-group difference of 0.24%, and a within-group change of 0.08% in the intervention group. Indeed, the selection of MSC as 0.25% is a controversial area, as most trials consider 0.5% as a minimal significant change. (One might consider a lower MSC in population-level studies of a self-management program; however, this is not the case in this patient-level trial.) Finally, while authors reference UKPDS to address clinical significance of their finding, it is not clearly acknowledged that the 0.08% reduction achieved in the trial is quite different from the 1% reduction achieved by UKPDS, and the subsequent reductions in complication and mortality. As in my previous review, I do not dispute these findings – they are promising! They must however be interpreted with caution and a cautious tone used in the abstract, results and discussion – these are very modest differences, the clinical significance of which is certainly debatable by researchers and clinicians alike. The authors should acknowledge this limitation in the Discussion.
--

REVIEWER	Chris Penfold Senior Research Associate, University of Bristol, UK
REVIEW RETURNED	21-Jun-2017

GENERAL COMMENTS	No further comments
---------------------

REVIEWER	Resmi Gupta Cincinnati Children's Hospital Medical Center Cincinnati, Ohio, USA
REVIEW RETURNED	20-Jun-2017

GENERAL COMMENTS	The authors provided clear responses to reviewer's comments and edited the manuscript to make it stronger , well constructed research article.
--

VERSION 2 – AUTHOR RESPONSE

We have added an additional bullet point to the strengths and limitations (editorial request), and altered the first paragraph of the discussion to reflect reviewer 2's comments.

As requested we have uploaded an unmarked copy, plus high resolution images for Figures 1 - 3.